# Proteomic Mapping of the Interactome of KRAS Mutants Identifies New Features of RAS Signalling Networks and the Mechanism of Action of Sotorasib

**DOI:** 10.3390/cancers15164141

**Published:** 2023-08-17

**Authors:** Aoife Nolan, Cinzia Raso, Walter Kolch, Alex von Kriegsheim, Kieran Wynne, David Matallanas

**Affiliations:** 1Systems Biology Ireland, School of Medicine, University College Dublin, Belfield, D04 V1W8 Dublin, Ireland; aoife.a.nolan@ucdconnect.ie (A.N.); cinzia.raso@gmail.com (C.R.); walter.kolch@ucd.ie (W.K.); alex.vonkriegsheim@igmm.ed.ac.uk (A.v.K.); kieran.wynne1@ucd.ie (K.W.); 2Conway Institute of Biomolecular and Biomedical Research, University College Dublin, D04 V1W8 Dublin, Ireland; 3Edinburgh Cancer Research UK Centre, MRC Institute of Genetics and Molecular Medicine, University of Edinburgh, Edinburgh EH4 2XU, UK

**Keywords:** KRAS, proteomics, JAK1, RADIL, Sotorasib, SOS1

## Abstract

**Simple Summary:**

Cancer is caused by changes in DNA called mutations that alter the way proteins work. We know that RAS proteins are one of the most commonly mutated proteins in cancer cells. These proteins work like a light switch and mutant RAS remain in the ON position sending signals to the cell to keep dividing when they should not. Until recently, it was thought that all RAS mutations were equal, but differences among these mutants have been identified. Here we used a technique called proteomics to decipher the differences among RAS mutants. We find that each mutant binds a different set of proteins and can regulate different signals. We also find that a clinically approved drug that inhibits one RAS mutant regulates the interaction of RAS proteins with other proteins. Our findings extend our knowledge of how the RAS mutants work, which can potentially be used to improve cancer treatments.

**Abstract:**

RAS proteins are key regulators of cell signalling and control different cell functions including cell proliferation, differentiation, and cell death. Point mutations in the genes of this family are common, particularly in *KRAS*. These mutations were thought to cause the constitutive activation of KRAS, but recent findings showed that some mutants can cycle between active and inactive states. This observation, together with the development of covalent KRASG12C inhibitors, has led to the arrival of KRAS inhibitors in the clinic. However, most patients develop resistance to these targeted therapies, and we lack effective treatments for other KRAS mutants. To accelerate the development of RAS targeting therapies, we need to fully characterise the molecular mechanisms governing KRAS signalling networks and determine what differentiates the signalling downstream of the KRAS mutants. Here we have used affinity purification mass-spectrometry proteomics to characterise the interactome of KRAS wild-type and three KRAS mutants. Bioinformatic analysis associated with experimental validation allows us to map the signalling network mediated by the different KRAS proteins. Using this approach, we characterised how the interactome of KRAS wild-type and mutants is regulated by the clinically approved KRASG12C inhibitor Sotorasib. In addition, we identified novel crosstalks between KRAS and its effector pathways including the AKT and JAK-STAT signalling modules.

## 1. Introduction

The three genes of the RAS family, *HRAS*, *KRAS*, and *NRAS*, code for four proteins and were described 40 years ago [1]. These proteins are members of the small GTPase superfamily, and they cycle between an inactive state when they are bound to GDP and an active state when they exchange GDP for GTP [2]. Inactivation occurs when GTP is hydrolysed to GDP in a reaction that can be catalysed by RAS proteins intrinsic GTPase activity. However, the activation switch of RAS proteins is tightly regulated by several proteins including the guanine nucleotide exchange factors (GEFs) that promote the exchange of GDP for GTP and the RAS GTPase activating proteins (GAPs) that accelerate the hydrolysis of GTP. Activated RAS-GTP binds to effector proteins that are ultimately responsible for the (patho)physiological functions mediated by RAS proteins including proliferation, migration, and differentiation [3]. Extensive work has identified more than 20 bona fide and putative RAS effectors [2]. However, despite intensive research, we lack a clear picture on how these proteins regulate their complex signalling networks.

Mutations of the RAS family of genes are common in cancer and they are driving oncogenes in some of the cancer types with higher prevalence [4]. In cancer cells, the three genes are mutated at different rates. KRAS is the most commonly mutated gene (85% of mutations), followed by NRAS (14%), while HRAS mutations are less frequent (~1%) [4]. Traditionally, hyperactivating mutations of RAS proteins were thought to lock RAS in the active GTP-bound state, due to the conformational changes in protein structure caused by mutations in RAS [1]. However, recent evidence has disproved this dogma as it has been shown that many KRAS mutants retain varying levels of GTPase activities [5,6,7,8,9]. The current view is that the amino acid substitution involved is the determining factor of KRAS mutant GTPase activity, and subsequent activation of downstream pathways [7]. Biochemical studies have determined that the KRASG13D mutant has the highest GTPase activity with a 14 times faster GDP–GTP exchange rate than wild-type KRAS (hence forth KRAS WT) and nine times slower GAP-mediated GTP hydrolysis rate [6,9]. In clear contrast, KRASG12V shows a 1.8-fold reduced GDP–GTP exchange rate but it is insensitive to GAP-mediated GTP hydrolysis [6]. However, the determination of the GTPase activity of the different KRAS mutants is not completed and conflicting results in a study performed in MCF10A breast cancer cells found that KRASG12D and KRASG13D mutants had similar GTPase activity to KRAS WT, but that KRASG12C and KRASG12V had higher GTP binding [7]. Importantly, the binding of many KRAS effectors is GTP-dependent, including the best characterised KRAS effectors RAF1, PI3K, RALGDS, and PLCε [10,11]. Thus, the fact that the different mutants are regulated by GTP/GDP exchange can indicate that different KRAS mutants can have different affinities for interacting proteins and some mutations can result in the formation of oncogenic neo-interactions with proteins that do not bind to KRAS WT. This hypothesis is supported by recent data showing that KRAS interaction affinity with the RAF RAS-binding domain (RBD) decreases, dependent on the mutation present at KRAS codons 12, 13, and 61 [9]. In this case, KRASG13D, KRASQ61L, KRASG12A, and KRASG12C mutants had only slightly lower interaction affinity with RAF-RBD compared with KRAS WT. Conversely, KRASG12V, KRASG12D, and KRASG12R mutants had significantly lower interaction affinity with RAF-RBD. Thus, different types of KRAS mutations can profoundly change the interaction with effectors and downstream signalling.

Importantly, in the last decade there has been a paradigm change in the way we approach the development of targeted therapies against RAS-driven tumours. For decades, RAS proteins were considered undruggable. The development of a series of covalent inhibitors specific for KRASG12C and the FDA approval of Sotorasib (AMG510) and Adagrasib demonstrate that we can develop effective RAS inhibitors, at least for some of the most common mutants [12]. Unfortunately, while the initial response rates to Sotorasib and Adagrasib are promising, ranging from 32% to 46%, respectively, in non-small cell lung cancer (NSCLC), resistance inevitably develops in these patients [13]. These drugs can also lead to severe side effects in the patients and more information is necessary to fully characterise their mechanism of action [14].

Traditionally, all KRAS point mutations observed in human tumours were considered to mediate their transforming potential through the same effector pathways. Importantly, increasing evidence shows that specific mutations are associated with different transforming potential and differential regulation of metastasis in various cell and animal models [7,15]. The existence of pathophysiological differences among the different mutant KRAS is further supported by clinical observations indicating that patients with mutant KRAS colorectal cancer may respond to targeted therapy depending on the type of point mutation they express [16]. In particular, KRASG13D but not KRASG12V tumours were shown to be sensitive to EGFR inhibition. These differences are probably related to the different GTPase activity of these mutants which likely results in different interactomes associated with specific KRAS mutants. For this reason, in the past few years, several groups have performed affinity purification mass-spectrometry (AP-MS) studies to characterise the specific interactome of several mutant KRAS [10]. Collectively, these studies clearly support the existence of specific mutant KRAS interactomes. 

Here, we extend these studies using AP-MS to compare the interactome of KRAS WT, KRASG12V, and KRASG13D in non-tumoral and tumoral cell lines. Using AP-MS we identify specific transient interactors and novel RAS interactors. We experimentally validated our findings, extending our studies to other KRAS mutants, demonstrating that KRAS mutants have different affinities for some of their interactors including SOS1. In light of these findings, we applied our AP-MS proteomic approach to characterise the effect of Sotorasib on the interactome of KRASG12C and KRAS WT. Our findings show a regulation of the KRASG12C interactome by this drug. Surprisingly, we also see a Sotorasib-mediated regulation of KRAS WT interacting proteins and an activation of AKT. Our analysis identified a novel interaction of KRASG12C with JAK1, with an unexpected regulation of KRAS protein expression by the JAK pathway. We also identified proteins of the AKT pathway which may be involved in the activation of AKT. Altogether, our results expand our understanding of KRAS signalling networks and provide mechanistic insights into why not all KRAS mutants are created equal. 

## 2. Materials and Methods

### 2.1. Cell Culture and Transfection

HEK293 (ATCC), RAS-less mouse embryonic fibroblasts (MEF) isogenic cell lines [17] (from the RAS initiative at Frederick National Laboratory), and HKe-3 (Shirasawa et al. 1993) cells were maintained in T75 flasks at 37 °C in 95% humidified air containing 5% CO_2_ in HEPA class 100 steri-cycle CO_2_ incubator (Thermo Electron Corporation, Vantaa Finland) and cultured in Dulbecco’s Modified Eagle Medium (DMEM) containing 10% Foetal Bovine Serum and 2 mM L glutamine (all from Gibco, Waltham, MA, USA). NIH3T3 (ATCC) cells were maintained at similar conditions and cultured in filtered DMEM containing 10% Calf Serum (Sigma-Aldrich, Burlington MA, USA) and 2 mM L-glutamine. Cells were transfected with Lipofectamine 2000 (Invitrogen, Waltham, MA, USA), according to the manufacturer’s instructions. All cells were validated by genome sequencing in the last 12 months. For stimulation with EGF (Roche, Basel, Switzerland), following 24 h transfection of HEK293 cells, cells were starved for 16 h prior to stimulation with 3 µM EGF for 5 and 10 min, and cells were then lysed. Cells were treated with 5 µM Sotorasib (MedChemExpress, Monmouth Junction, NJ, USA) and 5 µM Ruxolitinib (SelleckChem, Berlin, Germany) for 24 h in normal growth conditions prior to protein extraction by cell lysis. 

### 2.2. Immunoblotting, Antibodies and Reagents

Cell extract protein concentrations were measured by Pierce BCA assay (Invitrogen) as outlined by the manufacturer’s instruction. Extracts and immunoprecipitates were denaturalised and analysed by SDS-polyacrylamide gel electrophoresis and transferred to PVDF membranes (Millipore, Darmstadt, Germany), and blotted with commercial antibodies. Rabbit polyclonal RADIL (Proteintech, Manchester, UK), monoclonal anti-FLAG M2-HRP (MERK, Darmstadt, Germany), mouse monoclonal α-Tubulin antibody (TU-02; Santa Cruz, Dallas, TX, USA), rabbit polyclonal GAPDH (Cell Signaling, Danvers, MA, USA), goat polyclonal RIN1 (N-19; Santa Cruz), mouse monoclonal p-ERK (E-4; Santa Cruz), rabbit polyclonal ERK1 (C-16; Santa Cruz), rabbit polyclonal p-S473-AKT (Cell Signaling), rabbit polyclonal AKT1 (N-19, Santa Cruz), rabbit polyclonal Vinculin (Cell Signaling), mouse monoclonal BRAF (F-7; Santa Cruz), mouse monoclonal KRAS (F234; Santa Cruz), rabbit polyclonal MEK (12-B; Santa Cruz), rabbit polycloncal p-S217/221-MEK (Cell Signaling), mouse monoclonal panRAS (Ab-3; EMD Millipore, Germany), mouse monoclonal RAF1 (E-10; Santa Cruz), rabbit polyclonal p-S289/296/301-RAF1 (Cell Signaling), rabbit polyclonal SAPK/JNK (Cell Signaling), rabbit polyclonal p-T183/Y185-SAPK/JNK (Cell Signaling), rabbit polyclonal STAT5 (C-17; Santa Cruz), rabbit polyclonal p-Y705-STAT3 (Cell Signaling), rabbit polyclonal JAK1 (HR-785; Santa Cruz), rabbit polyclonal p-Y1034/1035-JAK1 (Cell Signaling), rabbit polyclonal Cleaved Caspase 3 (Cell Signaling), and mouse monoclonal AU5 (Covance, Burlington, NC, USA). 

### 2.3. Constructs and siRNA

Constructs encoding pCEFL-FLAG-KRAS WT, pCEFL-FLAG-KRASG13D, pCEFL-FLAG-KRASG12V, pCEFL-FLAG-KRASG12C, pCEFL-FLAG-KRASG12D, pCEFL-HA-NRASG12V, pCEFL-HA-KRAS WT, pCEFL-HA-KRASG13D, pCEFL-HA-KRASG12V, pCEFL-HA-KRASG12C, pCEFL-HA-KRASG12D, and pCEFL-AU5-SOS1 have been described before [18,19,20,21]. Small interfering RNAs (siRNAs) against RADIL human (SO-2776281G) and mouse (SO-2858456G) and non-targeted were from Dharmacon/Horizon (Lafayette, CO, USA). 

### 2.4. Immunoprecipitation (IP) and in Beads Tryptic Digestion

Cells were transfected with pCEFL-FLAG-KRAS-WT, -KRASG13D, -KRASG12V, -KRASG12C, -KRASG12D, and lysed after 48 h in 25 mM HEPES (pH7.5), 150 mM NaCl, 10 mM MgCl_2_, 1 mM EDTA (all Sigma-Aldrich), 1% NP-40 (Calbiochem, San Diego, CA, USA), protease, and phosphatase inhibitors (Roche). Cellular debris was removed by centrifugation at 20,000× *g* at 4 °C for 10 min. Baits were immunoprecipitated with anti-FLAG M2 conjugated agarose beads (Sigma-Aldrich). After 2 h incubation at 4 °C, immunoprecipitates were washed twice with lysis buffer containing 1% NP-40 and separated by SDS-PAGE. Blots were quantified using ImageJ version 1.50. For immunoprecipitation of endogenous RADIL the lysates were incubated with 10 µL Protein-G Sepharose beads (Sigma-Aldrich) with 2.6 µg/mL RADIL antibody at 4 °C for 2 h. Samples used in the MS analyses were immunoprecipitated with anti-FLAG M2 conjugated agarose beads, performed as above but after 2 h incubation at 4 °C the immunoprecipitates were digested with proteases as described before [22]. Briefly, immunoprecipitates were washed three times with lysis buffer containing 1% NP-40 and twice with lysis buffer that does not contain NP-40. Proteins were eluted from beads with 2 M Urea, 50 mM Tris-HCl (pH7.5) containing 5 µg/mL modified sequencing-grade trypsin (Promega, Madison, WI, USA) for 30 min. Proteins were digested in 2 M Urea, 50 mM Tris-HCl pH7.5, and 1 mM DTT (all from Sigma-Aldrich) containing 5 µg/mL modified sequencing-grade trypsin overnight at room temperature. Peptides were alkylated using iodoacetamide (5 mg/mL) and incubated in the dark for 30 min at room temperature. C18 Stage Tips were used to desalt samples, as described by [23] and analysed by mass spectrometry. For double digestion, proteins were denatured in 8 M Urea and reduced in 1 mM DTT for 30 min at room temperature. Proteins were alkylated in the dark for 30 min at room temperature with iodoacetamide (3 mM). Proteins were digested in 8 M Urea containing 3 ng/µL Lysyl Endopeptidase (Lys-C, Osaka, Wako, Japan) for 4 h at 37 °C. Samples were diluted to a final concentration of 2 M Urea and digested again overnight with 3 ng/µL modified sequencing grade trypsin. C18 Stage tips were used to desalt samples, as before, and analysed by mass spectrometry.

### 2.5. Mass Spectrometry Analysis of Affinity Purification-MS (AP-MS)

Experiment IP KRAS WT, KRASG12V, and KRASG13D: the samples were run on a ThermoScientific Q-Exactive mass spectrometer (ThermoFisher Scientific, Waltham, MA, USA) connected to Dionex Ultimate 3000 (RSLCnano, ThermoFisher, Waltham, MA, USA) chromatography system. Tryptic peptides were resuspended in 0.5% acetic acid, and 2% acetonitrile (both Sigma-Aldrich). Each sample was loaded onto a fused silica emitter, 75 μm ID, pulled using a laser puller (Sutter Instruments P2000, Novato, CA, USA), packed with Reprocil Pur C18 (1.9 μm, Germany) reverse phase media, and was separated by an increasing acetonitrile gradient over 120 min at a flow rate of 200 nL/min. The mass spectrometer was operated in positive ion mode with a capillary temperature of 320 °C, and with a potential of 2300 V applied to the frit. All data were acquired with the mass spectrometer operating in automatic data-dependent switching mode. A high resolution (70,000) MS scan (350–1600 *m*/*z*) was performed using the Q-Exactive to select the 12 most intense ions prior to MS/MS analysis using HCD.

Experiment IP Sotorasib treatment KRAS WT and KRASG12C: Samples were analysed on a Bruker timsTof Pro mass spectrometer (Bruker, Billerica, MA, USA) connected to an Evosep One liquid chromatography system (Eveosep, Odense, Denmark) or a Bruker nanoElute nanoflow chromatography system. In the case of the Evosep platform tryptic peptides were resuspended in 0.1% formic acid (Sigma-Aldrich) and each sample was loaded onto an Evosep tip. The Evosep tips were placed in position on the Evosep One, in a 96-tip box. The autosampler was configured to pick up each tip, and elute and separate the peptides using a set chromatography method (30 samples a day) [24]. For the Bruker nanoElute nano-LC chromatography system, each sample was loaded onto Acclaim PepMap C18 trap cartridge (0.3 mm inside diameter, 5 mm length, ThermoFisher Scientific) and then separated on an Aurora UHPLC column (25 cm × 75 μm ID, C18, 1.6 μm, Ionopticks, Fitzroy Australia) with an increasing acetonitrile gradient over 30 min at a flow rate of 250 nL/min [25]. The chromatography buffers used for both systems were Buffer A: 99.9% water, and 0.1% formic acid; and Buffer B: 99.9% acetonitrile, and 0.1% formic acid. All solvents used were LCMS grade. The mass spectrometer was operated in positive ion mode with a capillary voltage of between 1600 and 1800 V, dry gas flow of 3 L/min and a dry temperature of 180 °C. All data were acquired with the instrument operating in trapped ion mobility spectrometry (TIMS) mode. Trapped ions were selected for ms/ms using parallel accumulation serial fragmentation (PASEF). A scan range of (100–1700 *m*/*z*) was performed at a rate of 5 PASEF MS/MS frames to 1 MS scan with a cycle time of 1.03 s [26]. 

### 2.6. Analysis of AP-MS Data

The resulting mass spectra were analysed with MaxQuant [27] software, containing a built-in Andromeda search engine to identify the proteins from the UniProt HUMAN database [28] (release 2014_02). Protein interactions were quantified by LFQ Intensity. This consisted of 3 biological replicates and 2 technical replicates with 3 baits and empty vector controls in 2 cell lines. The raw data were searched against the Homo sapiens subset of the Uniprot Swissprot database (reviewed) using the search engine Maxquant (release 2.0.1.0) using specific parameters for trapped ion mobility spectra data-dependent acquisition (TIMS DDA). Each peptide used for protein identification met specific Maxquant parameters, i.e., only peptide scores that corresponded to a false discovery rate (FDR) of 0.01 were accepted from the Maxquant database search. The normalized protein intensity of each identified protein was used for label-free quantitation (LFQ) [29].

Contaminants and reverse peptides were removed. The statistical analysis was performed as previously reported [23,30]. In essence, the average protein interaction for each bait was measured by LFQ values. As the first step in the analysis, significant proteins were identified in comparison to the empty vector control. The ratio of protein abundance and Student’s *t*-test were used to select proteins significantly interacting with the bait proteins compared with the empty vector control (ratio > 2, *p*-value < 0.01). Specific proteins were pooled. For the experiment performed with HKe-3 cells described in Section 3.2 bait protein levels (KRAS) were assessed and normalised to remove potential protein interaction bias for one bait over the others. The ratio between conditions were calculated pairwise. To select proteins significantly interacting compared with other baits, the ratio of each protein, combined with Student’s *t*-test were used to identify proteins significantly interacting (ratio > 2 OR < 0.5, *p*-value < 0.01).

Cytoscape (version 3.7.2) was used to construct the interactomes of each KRAS isoform in both cell lines. String DB [31], REACTOME [32], and PANTHER databases [33] were used for gene ontology analysis, network reconstruction, and K-means cluster identification.

### 2.7. Transformation Assay

Transformation assays were performed as described by Aaronson et al. [34] and further explained in Herrero et al. [35]. Briefly, NIH3T3 cells were transfected with pCEFL-HA-KRAS WT, KRASG13D, KRASG12V, KRASG12C, KRASG12D, and/or 20 nM siRNA-RADIL. Cells were grown in culture conditions for at least two weeks with the media changed every 2–3 days. When macroscopic foci were visible, the cells were fixed with methanol for 5 min and stained with 5% Giemsa (in PBS) and the transformed foci were counted. The number of foci counted were quantified relative to 1 µL of DNA. Three biological replicates were quantified and averaged. 

### 2.8. Migration Assay

HKe-3 cells were transfected with pCEFL-FLAG-KRAS-WT, -KRASG13D, -KRASG12V, -KRASG12C, -KRASG12D, and +/− 4nM siRNA-RADIL; 24 h post-transfection, cells were counted and seeded into migration culture inserts (Ibidi, Grafelding, Germany). Culture inserts were removed 24 h post-seeding of cells. Cells were supplemented with DMEM containing 1% FBS and 1% L-glutamine. Pictures of the cell gap were taken at 0 h, 6 h and 24 h. The area of the gap was measured using ImageJ. Results were normalised to the 0 h time point for each condition, as outlined by Cappiello et al. [36] and percentage gap closure was quantified.

### 2.9. Survival Curves

To measure survival in HKe-3 we used MTS assays. To perform this, cells were transfected with pCEFL-FLAG-KRAS-WT, -KRASG13D, -KRASG12V, -KRASG12C, -KRASG12D, and/or 10 nM siRNA-RADIL. Cells were collected at the indicated times and MTS assay was performed as outlined by the manufacturer (Promega).

### 2.10. Statistics

Transformation assay: Number of foci were counted and quantified relative to 1µg DNA. Positive (upper) error bars are Max-Q3 (maximum [highest data point in the data set]—Q3 [third quartile, the median of the upper half of the dataset]), minus (lower) error bars are the error amount set to 100%. Upper (dark grey) boxes are Q3-Med (Q3 [third quartile, the median of the upper half of the dataset]—Med [median, the middle value in the dataset]). Lower light grey boxes are Med-Q1 (Med [median, the middle value in the dataset]—Q1 [first quartile, the median of the lower half of the dataset]). *p*-values were determined by two-tail, two-sample equal variance Student’s *t*-test; * *p* < 0.05, ** *p* < 0.01, *** *p* < 0.001; differences without *p*-values indicated were not significant.

Migration assay: Percentage cell gap was quantified using ImageJ and normalised to the 0 h time point gap as described by Cappiello et al. [36]. Error bars are +/− standard error of the mean (SEM) of three biological replicates. *p*-values were determined by two-tail, two-sample equal variance Student’s *t*-test; * *p* < 0.05, ** *p* < 0.01, *** *p* < 0.001; differences without *p*-values indicated were not significant.

Western blot quantification: Protein band intensity was quantified by ImageJ. Immunoprecipitates were normalised to the band intensity of bait proteins, phospho-proteins were normalised to band intensity of the corresponding total-protein levels. All samples were normalised to the control (KRAS WT) levels. *p*-values were determined by one-tail, two-sample equal variance Student’s *t*-test; * *p* < 0.05, ** *p* < 0.01, *** *p* < 0.001; differences without *p*-values indicated were not significant.

MTS cell survival assay: Three biological replicates with three technical replicates each. Percentage cell viability was normalised to the control (live cells). Error bars are +/− standard error of the mean (SEM) of three biological replicates. *p*-values were determined by two-tail, two-sample equal variance Student’s *t*-test; * *p* < 0.05; differences without *p*-values indicated were not significant. 

## 3. Results

### 3.1. KRAS Mutants Differentially Regulate the RAF and AKT Effector Pathways in HEK293 Cells

Different lines of evidence have shown that the specific point mutations of the KRAS oncogene are not equal [37,38,39,40,41]. To obtain a better understanding of the biochemical and functional differences among these proteins, we first tested the effects of the expression of KRAS WT and four of the most common oncogenic KRAS mutants, KRASG13D, KRASG12V, KRASG12C, and KRASG12D, on the regulation of ERK and AKT pathways in HEK293 cells. As expected, overexpression of KRAS WT caused a modest increase on ERK phosphorylation, while overexpression of the four KRAS mutants increased ERK phosphorylation (Figure 1A,B). KRASG12D induced phospho-ERK levels 12-fold compared with WT, followed by KRASG12C and KRASG13D with a 7.3- and 5.6-fold increase, respectively. KRASG12V was consistently less potent and induced ERK phosphorylation only 3-fold even though it was expressed at higher levels than KRASG13D. The effect on the levels of phospho-AKT were less striking, but KRASG12C and KRASG12D significantly induced phosphorylation of this kinase while expression of KRASG13D and KRASG12V had no effect (Figure 1A,C). Altogether, the results show that KRASG12C and KRASG12D have similar effects in the activation of ERK and AKT. KRASG13D does not activate the AKT pathway while KRASG12V has a lower effect in the activation of the ERK pathway than the other mutants and no effect on the activation of AKT. 

### 3.2. The KRAS Proteins Interactome Differs between Cell Types

We hypothesised that the results of the previous section might be explained, at least in part, by differences in the interactome of these mutants. To test this, we performed a proteomics screen of KRAS mutants interactomes. In order to facilitate the experimental analysis, we selected only two of these mutants, KRASG12V and KRASG13D. The rationale for this selection was that these two KRAS isoforms have the greater variation in the GTPase activity and differential response to EGFR inhibitors [7,16], which we reasoned may be due to the presence of very different interactors in their proteomes. We selected the human HEK-293 cells which are an immortalized cell line commonly used for interaction proteomics and has been used to explore RAS proteins interactomes [38,42,43]. Also, we were interested in determining if there are differences in the interactome of these mutants in different cell lines and, in particular, in a cancer type where KRAS is commonly mutated. In our experience, overexpression of KRAS mutants in cells that have mutations in KRAS or BRAF genes is extremely difficult as this seems to result in oncogene-induced stress. For this reason, in addition to HEK293 cells, we performed a proteomic screen in HKe-3 cells, a colorectal cancer cell line derived from HCT116 cells where the oncogenic KRASG13D allele was knocked out that tolerates overexpression of KRAS mutants and we used before to perform proteomics screens [44,45]. We transfected FLAG-tagged constructs of KRAS WT, KRASG12V, and KRASG13D into these cell types (Figure 2A). Following IP, we determined the specific interactomes associated with the different KRAS proteins by AP-MS. Using label-free quantitation (LFQ), we performed statistical analysis to check if there were significant changes in the interactome of the three KRAS proteins expressed in the experiments. A first comparison using a Venn diagram representation showed clear differences in the interactomes associated with KRAS proteins in both cell lines with only 12 proteins out the 450 specific interactors being present in KRAS IPs performed in both cell lines (Figure 2B and Appendix A). These results showed that there are very pronounced cell specific differences in the interactome of KRAS proteins. To continue the systematic analysis, we focused on the differences in the interactome for the three KRAS baits in each cell line.

In HEK293 cells, 1530 proteins were identified in the screening, of which 400 were specifically binding to at least one of the KRAS proteins (Figure 2B,C and Appendix A). Using the statistical analysis explained in the material and methods section, we observed remarkable variations on the number of proteins binding to each of the KRAS proteins. As expected, there were common interactors to KRAS WT, KRASG13D, and KRASG12V (193), and other proteins were interacting with two of them (Figure 2C). Conversely, there were 46 proteins exclusively interacting with KRAS WT. In the case of the KRAS mutants, we also identified proteins exclusively binding to one of them with 48 proteins shown as unique KRASG13D interactors and only 16 shown as specific for KRASG12V. Similar results were observed in HKe-3 cells, although with clear differences in the distribution of the interactome (Figure 2D and Appendix A). In HKe-3 cells, we identified significantly fewer proteins (713 proteins), of which 62 proteins were shown as specific KRAS interactors. As in HEK293 cells the three proteins expressed shared part of the interactome, but only 11 proteins were common interactors. The most remarkable observation in these cells was that KRASG13D interacted with 22 proteins, a much lower number than the 40 binders of KRAS WT and the 41 interactors of KRASG12V. Despite these differences, all KRAS isoforms have unique interactors, with KRAS WT having the highest number (13), followed by KRASG12V and KRASG13D (13 and 7, respectively). Altogether, these results show that there are clear differences in the interactomes of the three KRAS isoforms and that they show cell specific characteristics.

Next, we evaluated the quality of our experiments by focusing on the proteins identified in our experiments. Several known RAS interactors were found in the analysis of the proteomic interaction data including proteins that have been identified in previous proteomics screenings performed with other methods such as BioID and AP-MS, among them the RAS effectors RAF1, BRAF, NRAS, HRAS, RADIL, or mTOR [37,39,40,46,47] (Figure 2E–N and Appendix A). The high enrichment of known KRAS interactors in our datasets demonstrate the validity of our method to investigate the specific interactomes of the different KRAS isoforms. Moreover, LFQ allowed us to find changes of affinity between the different KRAS baits and specific proteins. For instance, RAF1, BRAF, and ARAF have higher affinity for KRAS mutants than for KRAS WT in HEK293 cells (Figure 2E,F,J,K and Appendix A). However, while BRAF and RAF1 have similar affinity for both KRAS mutants, ARAF shows a higher affinity for KRASG12V (Appendix A). Importantly, in HKe-3 cells we saw further differences in the interaction of KRAS proteins with RAF family proteins. ARAF was not identified, and we saw that both RAF1 and BRAF have much higher affinity for KRASG13D than for KRASG12V (Figure 2J,K) indicating that there are cell specific factors that regulate the interaction of the KRAS mutants with RAF family effectors. 

Other known interactors of KRAS observed in the analysis of the HEK293 cells included MLLT4, SOS1, RHOA, RHOC, HRAS, and NRAS. Interactors of KRAS observed in the HKe-3 cell line included RIN1, ROCK1, HRAS, and NRAS. Interestingly, but perhaps not surprisingly, we saw that similar to ARAF some of these known interactors were only detected in one of the cell lines. For instance, MLLT4 and SOS1 were specific to the KRAS interactome in HEK293 cells (Figure 2H and Appendix A). On the other hand, the RAS effector RIN1 was only identified in HKe-3 cell IPs and showed higher affinity for KRASG13D (Figure 2M). This disparity in the number of interactors may be due to the expression levels of these proteins in both cell lines or can be specific to certain cell types, tissues, or the mutational status of oncogenic cell lines. In the case of the KRAS-RIN1 interaction, RIN1 is not expressed in HEK293 cells according to the Human Protein Atlas [48], explaining why we do not see this interaction in HEK293. CAT, MAPK3, NRAS, and DOCK7, which were shown to interact with KRAS baits only in HKe-3 cells (Figure 2N, Appendix A), are expressed in both cell lines suggesting that these interactions are cell type-specific.

### 3.3. Bioinformatic Characterisation of KRAS Protein Signalling Networks Show Correlation between KRAS Protein Interactomes and Function

We used gene ontology (GO) and pathway reconstruction to broadly determine the physiological and pathophysiological relevance of our findings. Panther GO enrichment analysis [49] showed some cell-type-specific variations in biological processes mediated by the KRAS proteins as a whole. For example, transporter activity GO is overrepresented in HEK293 but not in HKe-3 while the developmental process is enriched in HKe-3 cells (Figure 3A). Differences are also shown within each cell type for the GO enriched for specific mutants. For instance, in HEK293 cells the KRASG12V and KRASG13D interactomes are significantly enriched in proteins involved in signalling processes and protein binding versus KRAS WT (Appendix A). However, little differences between KRASG13D and KRASG12V were observed. Clearer differences between KRASG12V and KRASG13D interactomes were shown in the HKe-3 cell line (Appendix A). In these cells, KRAS WT and KRASG12V showed similar percentages of proteins involved in cellular and metabolic processes compared with the KRASG13D mutant which had more proteins involved in adhesion and in catalytic activity (Appendix A). 

Next, to obtain a network view of the interactomes associated with the three KRAS proteins in the two cell lines we used StringDB [31]. In HEK293 cells, the network showed a high number of connections between KRAS (Appendix A, red halo), which is to be expected as the dataset includes some of the best-known interactors such as RAF1, BRAF, and FNTA. Representation of a subnetwork containing highly enriched functional clusters identified an overrepresentation of proteins that are involved in metabolic processes, mTOR signalling, and Ras GTPase binding and cell cycle (Figure 3B and Appendix A). In the case of HKe-3, the signalling network of specific KRAS interactors showed an enrichment of proteins involved in different aspects of RNA binding and metabolism (Figure 3C). Altogether, these results confirm that the interactomes of KRAS are cell-type-specific and different KRAS proteins regulate specific effector pathways and functions which is in line with findings in recently published interactome studies [10].

### 3.4. Experimental Validation Confirms That KRAS Proteins Interactomes Are Differentially Regulated by KRAS Point Mutation and Shows That RADIL Mediates KRAS-Dependent Migration in a Mutant Specific Manner

Several of the findings described above can be of special functional relevance. In particular, we were intrigued by the observation that SOS1 might have different affinities for KRASG12V and KRASG13D in HEK293 cells (Figure 2H, of note this interactor was not identified in HKe-3). To validate this finding, we performed IP experiments in HEK293 cells co-transfected with AU5-SOS1 and FLAG-KRAS constructs. Of note, SOS1 expression levels were consistently lower when KRAS WT was expressed in these cells than when we expressed KRAS mutant constructs indicating that KRAS WT overexpression can regulate SOS1 levels by yet uncharacterised mechanisms. Nevertheless, as shown in the proteomics experiment, SOS1 had a higher affinity for KRASG13D than for KRASG12V (Figure 4A). Interestingly, KRAS WT had the highest affinity for SOS1 than any of the KRAS mutants. This disparity might be explained by the fact that we immunoprecipitated SOS1 while in the screen we pulled down KRAS complexes. Additionally, we could see that KRASG12C has similar affinity for SOS1 as KRASG13D, while KRASG12D has a higher affinity for SOS1 and KRASG12V has the lowest affinity. These differential affinities of KRAS mutants for SOS1 might have important implications in tumours, as KRAS binds to SOS1 in the enzymatic pocket but can also regulate its function by binding to an allosteric regulatory pocket [50]. Unfortunately, our attempts to experimentally distinguish the contribution of the two binding sites were not conclusive.

We also confirmed that RIN1 expression is not detected in HEK293 cells, but it is expressed in HKe-3 cells where it binds better to KRASG13D than KRASG12V or KRAS WT (Figure 4B). Finally, we validated the differential interaction of RADIL with the three KRAS proteins used in the proteomics experiments in both cell lines. In this case, we did see a very weak interaction with KRAS WT while the two mutants had higher affinity, with KRASG13D having the highest affinity for RADIL. Moreover, our results showed that the differential affinity of RADIL for the KRAS mutant isoforms also extends to the mutants KRASG12C and KRASG12D, that bind this protein with lower affinity than KRASG13D with KRASG12D, showing the lowest affinity of all the mutants tested (Figure 4C,D). In order to test if activation of KRAS WT increases its interaction with RADIL, we treated HEK293 cells with EGF, which caused a clear increase in KRAS WT-RADIL interaction (Figure 4E). This is similar to recent observations showing that RADIL interaction is increased upon KRAS WT activation [43]. Importantly, when we performed these experiments, RADIL was not considered a KRAS interactor but since then, data from other interaction proteomic studies identified this protein as one of the main interactors in AP-MS experiments using different RAS proteins [37]. 

Our results confirm that RADIL is an important KRAS interactor protein that is differentially regulated by specific KRAS point mutations. Choi et al. showed that this protein mediates KRAS-dependent proliferation and focal adhesion formation [43] and, Kelly et al. saw that RADIL mediates KRAS-dependent migration [37]. We decided to extend the functional characterisation of this interactor with the different KRAS mutants through functional biological assays. We first tested the possible role of RADIL in KRAS-dependent cell viability by knocking down RADIL with specific siRNA in HKe-3 cells expressing the different KRAS mutants. Although RADIL knockdown by specific siRNA was efficient (Appendix A), neither expression of the different KRAS proteins nor downregulation of RADIL expression affected cell viability (Appendix A). Next, we tested the role of RADIL in KRAS-mediated transformation by performing a focus formation assay in NIH3T3 cells. We transfected the cells with KRAS WT and the 4 KRAS mutants and downregulated RADIL protein expression by co-transfection of RADIL siRNA. All KRAS mutants induced the formation of transformed foci, although with clear differences (Figure 4F and Appendix A). KRASG13D showed the lowest transformation effect similar to KRAS WT, while KRASG12V, KRASG12C, and KRASG12D produced 5–8 times more transformed foci than KRASG13D. This result is in line with previous data showing that KRASG13D has lower transformation efficacy [15,51]. Downregulation of RADIL did not show a statistically significant effect on the number of foci induced by the different mutants (Figure 4F and Appendix A). Finally, given the known involvement of RADIL in cell motility [43] we studied the role of RADIL in KRAS-dependent migration using a wound healing assay. Downregulation of RADIL alone did not have any effect on the migration of HKe-3 cells (Figure 4G and Appendix A). Surprisingly, KRASG12D was the only KRAS protein that caused an increase in the migration of these cells. Importantly, this effect is mediated by RADIL as its downregulation reduced KRASG12D-dependent migration. Remarkably, RADIL seems to be a downstream effector of another mutant, KRASG12V, which did not increase migration in these cells. In this case, downregulation of RADIL increased migration of KRASG12V-expressing cells. Thus, RADIL might be playing different roles in the migration of tumour cells that express specific KRAS point mutations. This result strengthens the idea that not all the KRAS mutants have the same functions.

### 3.5. Sotorasib Shows Off-Target Effects in Cells That Do Not Express KRASG12C

The previous results clearly showed that there are changes in the interactome of KRAS mutants that can be of relevance for the initiation, development, and evolution of KRAS tumours. In light of these results, we tested whether the mechanism of action of the newly approved KRASG12C inhibitors includes the specific regulation of KRASG12C effector pathways mediated by changes in the interactome. For this, we monitored the effects of the KRASG12C mutant inhibitor, Sotorasib, on the regulation of the best characterised RAS effector pathways, RAF and AKT, in cell lines that express different KRAS mutants. Sotorasib inhibits KRASG12C at nM concentrations but in different studies is used at a wide range of concentrations [14,52]. Importantly, different groups are using high concentrations of Sotorasib (5–10 µM) to generate resistant cell lines that can be used to characterise the mechanisms of resistance to this drug [53,54,55]. For this reason, we decided to use high concentrations of Sotorasib in this part of the study. We used RASless mouse embryonic fibroblasts (MEFs) generated by Drosten et al. [17] that express one of the KRAS mutants and none of the other RAS (HRAS, NRAS, and KRAS) genes. The advantage of these cell lines is that we avoid any unspecific regulation of RAS signalling effector pathways that might be induced by Sotorasib treatment. In addition to MEF-KRASG12C, we assessed the effect of Sotorasib in MEF-KRAS WT, KRASG12V, and KRASG12D cells should not be affected by treatment with this drug even at these high concentrations [55]. Sotorasib treatment of MEF-KRASG12C caused a band shift in this mutant due to the covalent binding of this drug. This band shift did not occur in the other KRAS proteins (Figure 5A and Appendix A). As expected, Sotorasib only inhibited activating ERK1/2 phosphorylation and ERK induced feedback phosphorylation of RAF1 in MEF-KRASG12C confirming the specificity of this drug for this mutant. Similar results were observed in HEK293 cells where we could see that Sotorasib inhibits KRASG12C-dependent activation of the three core kinases of the ERK pathway (Figure 5B). We also observed an increase in AKT phosphorylation, similar to what has been described before in cancer cells [56]. Surprisingly, Sotorasib upregulated phospho-AKT in all four MEF cell lines, especially in MEF cells expressing KRASG12D (Figure 5A and Appendix A). The results clearly indicate that Sotorasib must have other targets that regulate AKT activation independent of KRASG12C. Interestingly, these targets seem cell-type-specific, as the induction of pAKT by Sotorasib was observed in MEFs and HKe-3 (Appendix A), but not in HEK293 cells (Figure 5B and Appendix A). As pAKT is constitutively upregulated in Sotorasib-resistant lung and pancreatic cancer cell lines [57], the off-target activation of AKT may be relevant for the development of resistance to Sotorasib.

### 3.6. Sotorasib Regulates the KRASG12C Interactome and Has Unspecific Effects on the KRAS WT Interactome

To characterise if Sotorasib’s mechanism of action involves changes in the KRASG12C interactome, we used the AP-MS proteomic approach described above. Unfortunately, MEF-KRASG12C cells cannot be used for AP-MS experiments, as the KRASG12C protein is not tagged and there are no suitable antibodies available for immunoprecipitating endogenous RAS and its effector proteins. Therefore, we used HEK293 cells transfected with FLAG-KRASG12C or FLAG-KRAS WT for comparison. Despite the fact that we immunoprecipitated similar amounts of KRAS WT and KRASG12C and Sotorasib treatment does not affect their protein expression, we saw lower level of KRASG12C in AP-MS (Figure 6A,B and Appendix A). This suggested that KRASG12C structure and Sotorasib binding interfered with the trypsin digestion of FLAG-KRASG12C that is part of the sample preparation for MS. It is well-known that some protein conformations can be resistant to trypsin digestion [58], and it seems that Sotorasib can induce such a protease-resistant conformation in KRASG12C. To test this, we performed double digestion with trypsin and LysC, which allowed the full digestion of untreated FLAG-KRASG12C and improved the digestion of Sotorasib-bound KRASG12C (Appendix A). The partial trypsin resistance of KRASG12C, however, did not affect the digestion of interacting proteins co-precipitating with FLAG-KRASG12C (Appendix A). Hence, we used standard trypsin digestion in the AP-MS experiments to study the effect of Sotorasib on its interactome, as the dataset was of better quality. 

Comparing the interactomes of FLAG-KRAS WT and FLAG-KRASG12C plus/minus Sotorasib treatment showed that 41 proteins were exclusively bound to KRASG12C and 110 interacted only with KRAS WT (Figure 6C). Of the 246 proteins that are specifically interacting with KRASG12C, 43 are exclusively binding when the cells are treated with Sotorasib, and 76 proteins are specifically interacting with KRASG12C in untreated cells (Figure 6C and Appendix A). Remarkably, we also observed that Sotorasib impacts the composition of the KRAS WT interactome; 55 proteins are only present in Sotorasib-treated cells and 39 in untreated cells. Additionally, we could see clear regulation of proteins that are interacting with both baits in treated and untreated conditions (Figure 6D,E and Appendix A) including the best-known interactors of KRAS, RAF1, and BRAF. Thus, our screen shows that Sotorasib changed both the KRASG12C and KRAS WT interactomes. In particular, the drug abolished the strong interactions of RAF1 and BRAF with KRASG12C, which explains the inhibition of ERK activation (Figure 6D). The binding to KRAS WT was much weaker and was not affected by Sotorasib. MLLT4 (Afadin), RADIL, and RAP1GDS showed a similar behaviour (Figure 6D and Appendix A). Both MLLT4 and RADIL are bona fide RAS interactors that bind to GTP-loaded RAS [41,43]. Therefore, this pattern of binding and inhibition by Sotorasib is expected. Interestingly, Sotorasib also reduced the strong binding of NRAS to KRAS WT and the weaker binding to KRASG12C (Figure 6E). These interactions may rely on activation, i.e., GTP-dependent, but not mediated through the effector domain. These results suggest that Sotorasib interferes with RAS oligomerization and nanoclustering, which are important for RAS function [59]. Binding to LAMTOR3, FYN, and STOM, which was stronger in KRAS WT, was increased by Sotorasib treatment in KRASG12C (Figure 6E and Appendix A). Interestingly, Sotorasib increased the binding to some proteins. For instance, binding to JAK1 was increased in both KRAS WT and KRASG12C (Figure 6E and Appendix A). Altogether, these results suggest indirect or off-target effects, which can contribute to the Sotorasib mode of action as they affect signalling molecules that play important roles in malignant transformation.

### 3.7. Sotorasib Regulates KRASG12C and WT Signalling Networks

We continued our analysis to obtain a systematic view of the Sotorasib-induced changes on the specific interactomes of KRASG12C and KRAS WT. First, we reconstructed the interaction network of proteins that are binding to each bait using the STRING database (Appendix A). The networks are extremely complex and show a high degree of connection among the nodes, which complicates the identification and representation of functionally relevant findings. For this reason, we represented the sub-network of the proteins that are changing their interaction with RAS upon Sotorasib treatment. In the case of KRASG12C, functional analysis confirms that there is an enrichment of proteins that are part of the AKT and ERK signalling pathways confirming the central role of these pathways in the mechanism of action of Sotorasib (Figure 7A). The changes of interaction of these nodes likely have an effect in some of the proteins that are part of these pathways but do not change their interaction with KRASG12C in our screen including PIK3R1/2 (the regulatory subunits of PI3K) or ILK. Interestingly, this analysis shows an increase in interaction with proteins involved in the functions of the lysosome including STOM, LAMTOR3, and several members of the ATPV6 complex (grey area Figure 7A). An association of KRAS4B with ATPV6 and lysosomal localisation was described before [38]. Thus, our data indicate that Sotorasib induces the localisation of KRAS in the lysosome. Intriguingly, a similar analysis of Sotorasib-regulated KRAS WT network identified a decrease in interaction of this protein with the ATPV6 complex indicating that Sotorasib prevents KRAS WT localisation to the lysosome (Figure 7B). Conversely, proteins involved in RNA recognition motif have an increased binding to KRAS WT treated with Sotorasib. Finally, JAK1 interaction with KRASG12C and KRAS WT is increased in cells treated with Sotorasib confirming that this drug promotes the interaction of this kinase with KRAS proteins. Altogether, these analyses showed that Sotorasib regulates the KRASG12C interactome but also has clear effect in the regulation of the KRAS WT interactome. 

Although Sotorasib is approved as a KRASG12C-specific inhibitor, different groups have identified other proteins that bind this drug and we reasoned that some of these targets might also be part of the KRAS interactome [14,52,60,61]. To explore this, we used information from one of the studies that have identified proteins that are targeted by Sotorasib using an MS-based approach [14]. This study identified 183 proteins that bind Sotorasib and using a Venn diagram analysis we saw that 15 Sotorasib target proteins were identified in our screen as part of the interactome of KRAS proteins (Figure 7C). Importantly, these proteins form a highly connected network and include proteins from the cytoskeleton including VIM, ACTB, and TUB1A/B; and the member of the 14-3-3 proteins YWHAQ which are key regulators of KRAS downstream effectors (Figure 7D). These data indicate that at least some of the changes in KRASG12C and KRAS WT interactomes induced by Sotorasib treatment might be caused by Sotorasib directly targeting protein interactors of KRAS. 

### 3.8. Validation Experiments Show a New Sotorasib-Regulated Crosstalk of RAS Signalling with JAK1

To experimentally validate our proteomics results, we performed KRAS immunoprecipitation assays and monitored the dynamics of the interaction of KRAS WT and KRASG12C and several of the proteins shown to be changing in the proteomics analysis. We confirmed that RADIL, RAF1, and BRAF interaction with KRASG12C was severely disrupted by Sotorasib in HEK293 cells (Figure 8A). Similar results were observed in HKe-3 cells for the interaction of KRASG12C with RADIL and RIN1 which are also downregulated by Sotorasib (Figure 8B). These validation experiments confirmed that our AP-MS method accurately identified changes induced by Sotorasib in the interactome of these KRAS proteins. They may differentially regulate its effector signalling pathways and therefore need to be understood to fully characterise the full mode of action of Sotorasib. 

One of the KRAS effector pathways that our results indicated to be modulated by Sotorasib is the JNK stress pathway, which was shown to be regulated in KRASG12C-expressing HEK293 cells. Indeed, we observed a clear KRASG12C-dependent activation of JNK that was completely abolished by Sotorasib treatment (Figure 8C). JAK1 signalling activation by KRASG12C induced phosphorylation of the STAT3, which was prevented by Sotorasib treatment (Figure 8C). To the best of our knowledge, this is the first demonstration that JAK1-STAT signalling may be a KRAS effector pathway, and that this pathway is modulated by KRASG12C specific inhibitors. 

JAK1 has been shown to regulate AKT1 in some cases [62], and we wondered whether JAK1 activation is related to Sotorasib-induced AKT activation. We tested this using Ruxolitinib, a JAK1/2 inhibitor used in the clinic [63], alone, or in combination with Sotorasib. In MEF-KRASG12C cells, there was a much lower level of STAT3 phosphorylation than in MEF-KRAS WT, but in both cell lines, addition of the JAK inhibitor clearly decreased the activation of STAT3 (Figure 8D). No changes in AKT1 phosphorylation were observed upon Sotorasib and Ruxolitinib combination treatment in MEF-KRASG12C cells indicating that JAK1 is not responsible for the Sotorasib-induced AKT activation in this cell line. In the case of MEF-KRAS WT, combination of Sotorasib and Ruxolitinib induced an increase in AKT phosphorylation indicating that the crosstalk of this protein with JAK is different to the KRASG12C-JAK1 crosstalk (Figure 8D). Similarly, we observed an increase in ERK1/2 phosphorylation in MEF-KRAS WT cells treated with Ruxolitinib, while the same treatment did not have any effect in ERK1/2 activation in MEF-KRASG12C. Finally, we observed an increase in caspase three cleavage in cells treated with the combination of KRASG12C and JAK1/2 inhibitors in MEF-KRAS WT but not in cells treated with either one of the treatments. In MEF-KRASG12C caspase cleavage was higher in all treatment conditions, indicating that there is an increase in apoptotic signalling in these cells. Finally, an important observation from these treatments was a conspicuous increase in the expression of KRAS WT and KRASG12C in cells treated with Ruxolitinib. This unexpected observation indicates that JAK signalling has an important role in the regulation of KRAS protein expression. Altogether, these results validate the findings from our proteomic screening and extend the characterisation of Sotorasib’s mechanism of action.

## 4. Discussion

Our study aimed to extend our characterisation of KRAS-interacting proteins and mapping the signalling networks regulated by KRAS WT and oncogenic mutants. The initial proteomics screen clearly showed that KRAS WT and two of the most common KRAS mutants, KRASG12V and KRASG13D, interact with different affinities to specific binding proteins. These results also show that the interactome of these proteins varies in the three cell lines that we used for the study. These findings are in line with several works that have used different MS-based proteomics approaches to identify the interactome of different RAS proteins [10,37,38,39,40,41,46,47,64]. One of the challenges of using AP-MS approaches is that some of the protocols used might not be suitable to identify transient interactions. To overcome this problem, several of the proteomic screens performed so far have used BioID [5,64], which potentially can identify transient and weak interactions of proteins within a ~10 nm radius of the bait [65]. However, our more direct approach clearly identifies interactors that have transient interaction with KRAS including the main effectors RAF1, BRAF, and PI3K. Moreover, we can quantify differences in affinities of a significant number of known and newly identified KRAS interactors including the best characterised GEF of the RAS family SOS1 [2] in HEK293 cells which, to the best of our knowledge, has not been identified using any other MS-based approaches reported to date. In the proteomics dataset SOS1 was shown to have much higher affinity for KRASG13D than KRASG12V and experimentally we show that KRASG12D and KRASG13D have the highest affinity for SOS1 while KRASG12V has very low affinity. This is an unexpected observation as KRASG13D was shown to be less dependent on SOS1 for activation than KRASG12V [9]. Although the functional relevance of this finding needs further studies, it is possible that the mutants with higher affinity can be binding to an allosteric pocket described by Bar Sagi’s group which regulates a positive feedback loop that has been proposed to promote tumorigenesis by activating RAS WT isoforms [50]. This might explain the higher level of activation of ERK1/2 when we overexpress KRASG13D.

Our method is also simpler to perform, as it does not rely on the expression of biotin ligase fusion proteins. It must be noted that a disadvantage of both methods is that it is not possible to differentiate between direct and indirect interactors of KRAS which requires further experimental validation or the use of crosslinking protocols suitable for AP-MS proteomics. Nevertheless, our approach is clearly suited to identify dynamic changes in KRAS protein interactomes and shows a great sensitivity to identify true interactors. This is extensively confirmed experimentally with our validation of the interaction of SOS1, RIN1, and RADIL with KRAS WT, KRASG12V, and KRASG13D. Our characterisation of the specific interactomes associated with KRAS mutants has been extended by including in our validation experiments two additional KRAS isoforms, KRASG12C and KRASG12D, which also show differential binding to several of the proteins studied. Hence, the work presented here elucidates a complex signalling network that shows common and specific modules for each KRAS isoform tested and further contributes to identify the growing number of RAS effector pathways. It must be noted that overexpression of RAS proteins and in particular KRAS mutant proteins at similar levels is sometimes challenging as shown in several of our experiments and in the literature [35,66]. Remarkably, this was also shown in the MEFs cells that express different levels of each KRAS mutant. This might be due to activation of pro-apoptotic signals by some mutants [67], rare codon stabilisation of KRAS transcripts [68], or existence of expression sweet spots [69]. To take these variations into account and minimise overinterpretation, we made sure that in all the immunoprecipitation experiments we have similar levels of KRAS proteins and quantified our results. Nevertheless, the findings further support the idea that not all KRAS mutations are equal, and that a given point mutation may be associated with specific pathological effects in tumour cells’ signalling networks. Overall, the results from our study must be considered in conjunction with previous reports that focus on mapping RAS protein interactomes and highlight the need to extend these screens to identify cell specific proteomes of all RAS isoforms. 

Importantly, our work goes further than previous proteomics screens as, for the first time, we characterised the changes caused by Sotorasib in the KRAS WT and KRASG12C interactomes. This KRASG12C-specific inhibitor is already approved for the treatment of lung cancer but resistance to treatment develops quickly [13]. Additionally, unspecific effects have been reported and it has also been shown to have effects in cell lines regardless of KRAS mutation status [70,71]. Reports also indicate that several patients develop severe side effects which indicate that at therapeutic concentration the drug has a systemic impact affecting cells that do not express KRASG12C [13]. One important consideration is that we used very high concentrations of Sotorasib in our study and some of the changes in KRASG12C and KRAS WT interactomes that we see in our screen might not occur at nM concentrations of Sotorasib. 

Our screen helps delineate the mechanism of action of this drug and can explain some of the side effects of Sotorasib. With respect to the mechanism of action, the dynamic changes shown in AP-MS confirms that it requires the regulation of the interactions of this mutant with effector proteins, something that was shown already for RAF1 and BRAF [37]. As expected, the interaction of these two bona fide effectors is prevented by Sotorasib treatment in our experiments, and this is accompanied by inhibition of ERK1/2 phosphorylation. Interestingly, we saw that Sotorasib also promotes the interaction of KRASG12C with other known KRAS regulators, such as members of the SRC family, i.e., LYN and FYN. With respect to the possible off-target effect of Sotorasib we also characterized an unexpected modulation of the KRAS WT interactome. These experiments indicated that Sotorasib can regulate the interactome of KRAS WT in a way that seems significant but needs further investigations. 

Additionally, we consistently observed a cell-type-specific increase in activating AKT phosphorylation in MEFs and HKe-3, but not HEK293. Activation of AKT upon Sotorasib treatment has been associated with resistance to this treatment in patients and is reported in several cell lines [13,57,71,72]. Importantly, our AP-MS approach identified clear changes in the interactomes of both KRAS WT and KRASG12C that potentially can explain the Sotorasib-induced AKT activation. Candidates for mediating this effect detected in our screen include PIK3R1/2, G-coupled receptor proteins, and ILK, which might contribute to the unspecific activation of AKT in these cells. It is worth noting our analysis shows that some of the changes shown in the interactomes of KRASG12C and KRAS WT upon Sotorasib treatment can be explained by the direct effect of Sotorasib on some of the proteins identified which are shown to be targeted by this drug in previous studies (i.e., ACTB, VIM or RAN) [14].

One important new finding from our study is the identification of a Sotorasib-regulated interaction between KRAS and JAK1 that has not been highlighted before. Reassuringly, it must be noted that a recent proteomics screen performed by the Kiel group using a similar protocol also pulled down JAK1 and STAT3 as specific interacting proteins of KRAS WT, KRASG12C, KRASG12V, and KRASG12D [47]. In our AP-MS proteomics experiment, we see that Sotorasib increases JAK1-KRAS interaction and at the molecular level we demonstrate that Sotorasib causes an inhibition of STAT3 phosphorylation in HEK293 cells but not in MEF cells that express only one of the KRAS proteins. This finding is particularly intriguing as it indicates that JAK1 might be specifically interacting with KRAS bound to GDP and this interaction prevents JAK1-dependent activation of STAT signalling. This new connection between KRAS and JAK-STAT signalling might be of clinical relevance as JAK1 inhibition has been shown to reduce murine KRAS mutant adenocarcinoma progression [63]. We show that the clinically approved JAK1/2 inhibitor Ruxolitinib increases caspase three cleavage in MEF-KRASG12C cells but not in MEF-KRAS WT. Remarkably, this experiment led to the unexpected observation that JAK1 inhibition alone causes a significant increase in expression of both KRAS WT and KRASG12C. Taking into account evidence from clinical trials that show an effect of JAK1 inhibitors in EGFR and possible KRAS mutant tumours (NCT02155465, NCT02145637, NCT02917993, and NCT03450330) [63], this serendipitous finding might be of relevance to explain a possible mechanism of action of JAK inhibitors in cancer. One possibility is that JAK1 inhibitors, among other mechanisms, might induce high levels of mutant KRAS expression that leads to the activation of the apoptotic pathways mediated by this RAS isoform [2,73]. It must be noted that this is an acute increase in KRAS expression, which might have very different effects than KRAS overexpression caused by gene amplification that has been associated with resistance to EGFR and KRAS targeted therapies in colorectal cancer cell lines [72]. Finally, our study extends the basic knowledge of the known crosstalk between the JAK and KRAS pathways. The role of JAK-STAT module in the RAS signalling network seems to be at different levels as exemplified by a recent study from Baccarini’s group showing a RAF1 kinase-independent regulation activation of STAT3 signalling in CRC [74]. It would be interesting to test in future studies if Sotorasib can somehow regulate this effect. 

## 5. Conclusions

In summary, this work extends our knowledge about the variety of functional pathways that KRAS proteins are mediating and confirms that the interactome of different KRAS mutants show important differences. Some of these new findings might prove important to explain the differences among KRAS mutant proteins in the promotion of cell transformation, tumour development, drug sensitivity, and maintenance of human tumours [1,2,4,16]. We also show evidence that performing proteomics screens is important to characterise the mechanisms of action of the recent FDA approved RAS targeting drugs and the new ones in drug development. Ultimately, determining if a RAS inhibitor prevents the interaction and/or regulation of specific RAS interactors can predict the mechanism of resistance that can hamper the effect of these novel therapies.

## Figures and Tables

**Figure 1 cancers-15-04141-f001:**
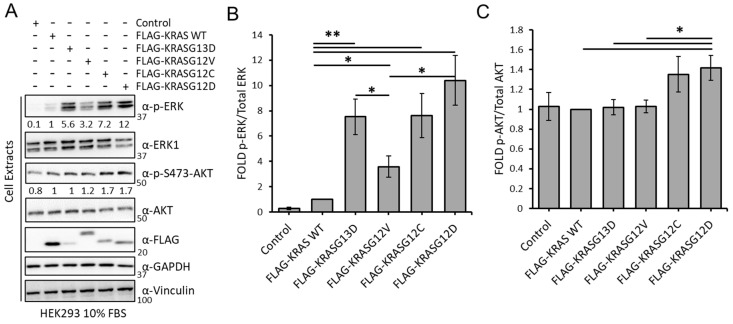
(**A**) HEK293 cells transfected with the indicated constructs were lysed after 48 h. Lysates were Western blotted with indicated antibodies. The whole Western blot figure can be found in Appendix A Original Blots and quantification for Figure 1A. Numbers indicate fold of phosphorylation of the indicated proteins normalised to the total protein. (**B**) graph shows the fold of phospho-ERK normalised to ERK expression (*n* = 3). Error bars show standard deviation. * *p* value < 0.05; ** *p* value < 0.01. (**C**) graph shows the fold of phospho-AKT normalized to AKT expression (*n* = 3). Error bars show standard error of the mean (SEM). * *p* value < 0.05; ** *p* value < 0.01.

**Figure 2 cancers-15-04141-f002:**
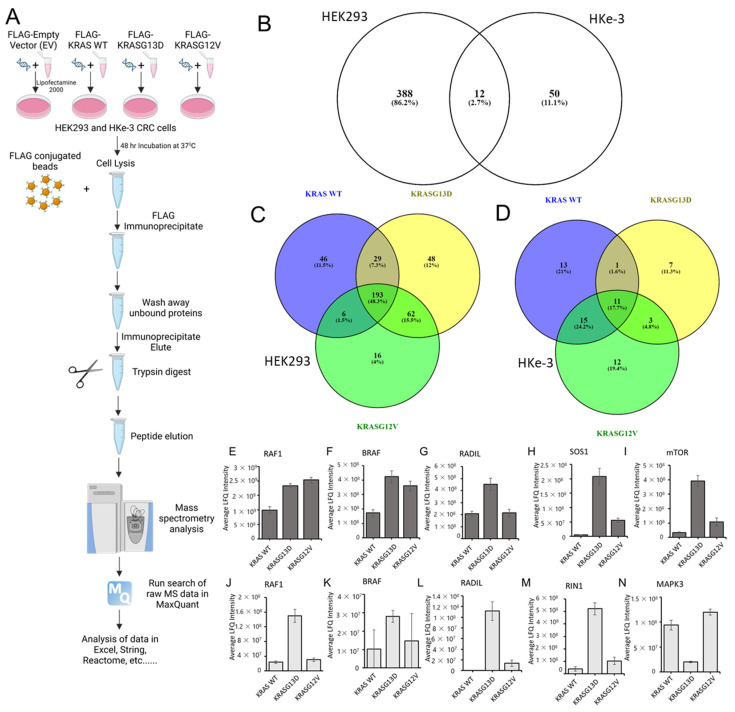
(**A**) Graphical representation of the mass spectrometry affinity purification protocol for KRAS WT, KRASG13D, and KRASG12V in HEK293 and HKe-3 cell lines. Cells were transfected with FLAG-tagged KRAS WT, G13D, and G12V DNA constructs. Cells were lysed after 48 h and incubated with FLAG-M2 conjugated to agarose. Following washes of the beads with lysis buffer, samples were prepared for trypsin digestion. Mass spectrometer protein searches of the raw data were conducted using Max Quant to quantify protein abundance associated with each bait. LFQ intensity of the protein for each condition was used to identify specific KRAS interactors in both cell lines. Created with BioRender.com. (**B**) Venn diagram representing the number of specific protein interactions binding to KRAS proteins (sorted by gene name) in HEK293 and HKe-3 cell lines (**C**) Venn diagram representing the specific interactors of the indicated KRAS proteins identified in AP-MS proteomics in HEK293 and (**D**) HKe-3 cell lines. (**E**–**I**) graphs show the average LFQ in HEK293 cells of the indicated proteins in immunoprecipitates of KRAS WT, KRASG13D, or KRASG12V as indicated (*n* = 3). Error bars show SEM. (**J**–**N**) graphs show the average LFQ in HKe-3 cells of the indicated proteins in immunoprecipitates of KRAS WT, KRASG13D or KRASG12V as indicated (*n* = 3). Error bars show SEM.

**Figure 3 cancers-15-04141-f003:**
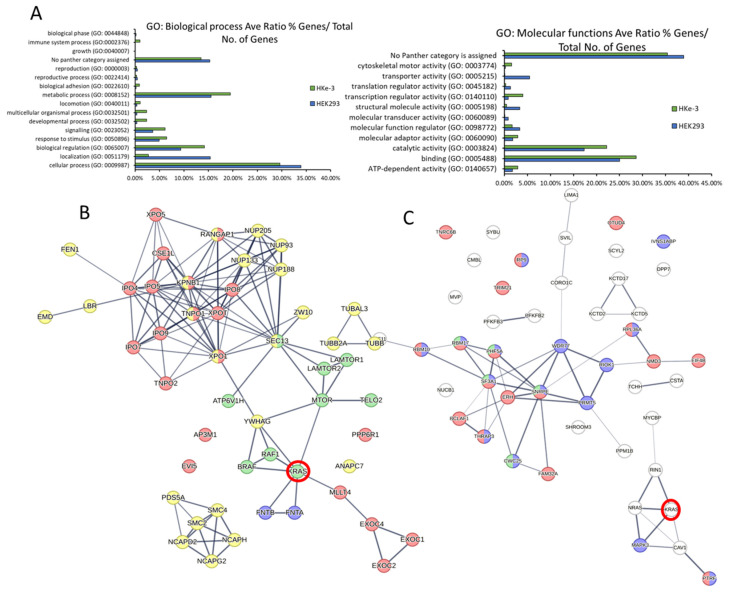
(**A**) The specific KRAS interactomes in HEK293 and HKe-3 cells identified by AP-MS were input into the Panther analysis tool. Graphs show representation of GO terms enriched for biological processes (left) and molecular functions (right). (**B**) Pathway reconstruction and cluster analysis performed with StringDB show HEK293 specific protein interactors of KRAS that are exclusive to the HEK293 cell line that are part of functional clusters. Red halo indicates the KRAS node. Edges represent confidence, line thickness indicates the strength of data support. Proteins coloured red are involved in RAS GTPase binding. Proteins coloured green are involved in mTOR signalling pathway. Yellow proteins are involved in Cell cycle-mitotic. Blue are proteins of the farnesyltransferase complex. (**C**) Specific protein interactors of KRAS that are exclusive to the HKe-3 cell line were reconstructed using StringDB. Red halo indicates the KRAS node. Edges represent confidence, line thickness indicates the strength of data support. Proteins coloured red are involved in RNA binding; proteins coloured blue are involved in RNA metabolic processing; green proteins are involved with the spliceosome.

**Figure 4 cancers-15-04141-f004:**
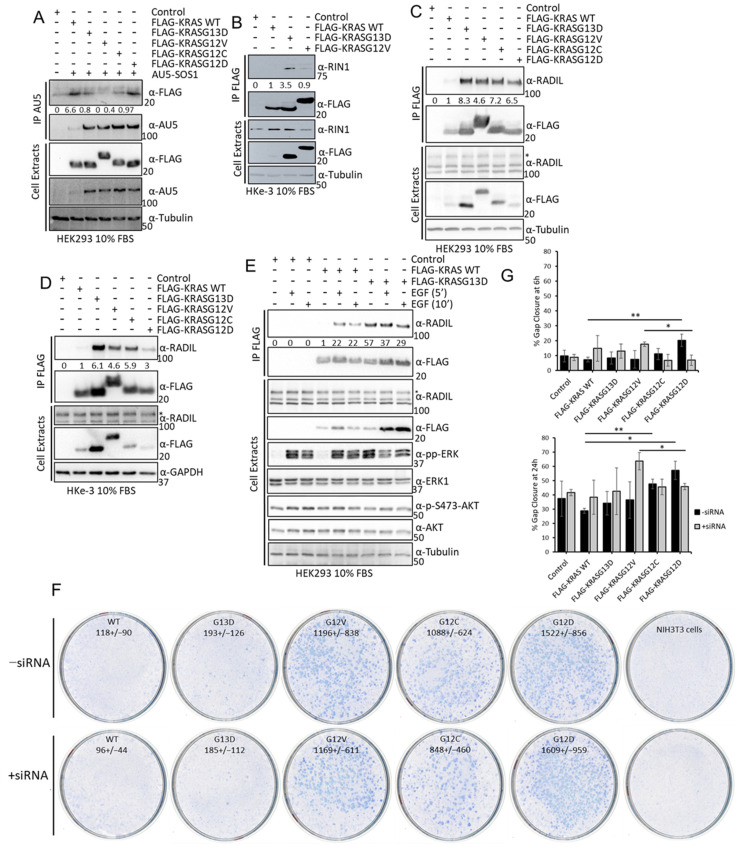
(**A**) HEK293 cells were transfected with the indicated FLAG-tagged KRAS proteins and AU5-SOS1. Cells were lysed after 48 h and IP of AU5-SOS1 was performed using the specific antibody. IPs and cell extracts were blotted with the indicated antibodies. (**B**) HKe-3 cells were transfected with empty vector, FLAG-KRAS WT, -KRASG12V, or -KRASG13D. Cell lysates were incubated with anti-FLAG antibody and immunoprecipitates were blotted with the indicated antibodies. Band intensities were measured using ImageJ and numbers show fold changes of the ratio RIN1/FLAG intensity in the IPs. (**C**) HEK293 cells or HKe-3 cells (**D**) were transfected with empty vector or the indicated KRAS plasmids and lysates were IP using FLAG antibody and blotted with the indicated antibodies. Numbers show fold changes of RADIL/FLAG intensities as measured using ImageJ. * indicates specific RADIL band in the cell extracts. (**E**) HEK293 cells were transfected with empty vector, FLAG-KRAS WT or –KRASG13D. Following 16 h serum deprivation cells were treated with EGF (3 μM) for the indicated times and lysed 48 h after transfection. FLAG IP was performed with the cell extracts followed by Western blotting with the indicated antibodies. Numbers show fold change in RADIL interaction with FLAG-KRAS proteins. Band intensity was measured using ImageJ. The whole Western blot figures can be found in Appendix A Original Blots and quantification for Figure 4. (**F**) NIH3T3 were transfected with 200 ng of FLAG-KRAS WT (WT), -KRASG13D (G13D), -KRASG12V (G12V), -KRASG12C (G12C), -KRASG12D (G12D), or empty vector and/or RADIL siRNA (20 nM); 14 days after, transfection plates were fixed and stained with Giemsa and macroscopic foci were counted. Numbers show average number of foci per 1 ug DNA +/− standard deviation (SD) (*n* = 3). (**G**) Cells transfected as in A were grown to confluence in a plate with a rubber stopper. Stoppers were removed and images were taken at the indicated times. Percentage gap closure was quantified using ImageJ and results were normalised to 0 h time point measurements [36]. Error bars are SEM. ** *p* value < 0.01, * *p* value < 0.05.

**Figure 5 cancers-15-04141-f005:**
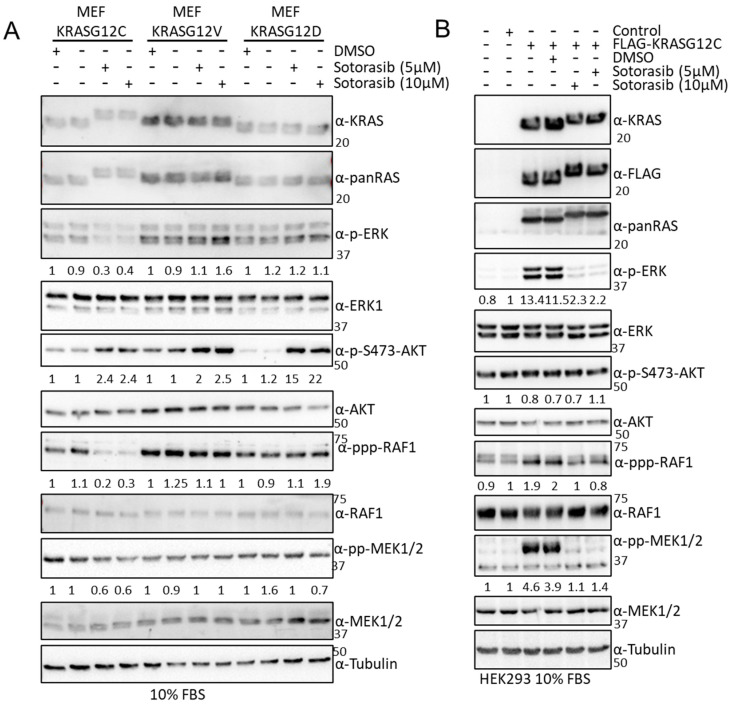
(**A**) MEF-KRASG12C, -KRASG12V, and –KRASG12D were treated with the indicated amount of Sotorasib for 24 h. Cells were lysed, and the indicated proteins were blotted using the specific antibodies. Numbers show fold changes of phosphorylation of the indicated proteins with respect to their total level of expression. Intensity was measured using ImageJ 1.5. (**B**) HEK293 cells were transfected with the indicated constructs; 24 h after transfection the cells were treated with the indicated concentrations of Sotorasib for 24 h. Cell lysates were blotted with the indicated. Numbers show fold changes of phosphorylation of the indicated proteins with respect to their total level of expression. Intensity was measured using ImageJ. Experiments were repeated 3 times. The whole Western blot figure can be found in Appendix A Original Blots and quantification for Figure 5.

**Figure 6 cancers-15-04141-f006:**
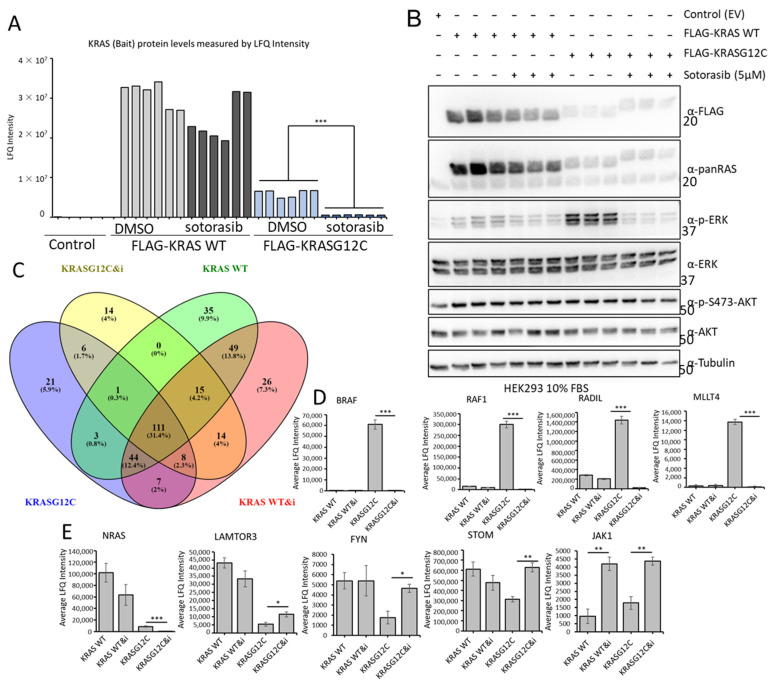
(**A**) HEK293 cells were transfected with the indicated constructs and 24 h later they were treated with Sotorasib (5 μM) or DMSO for 24 h. FLAG IP were tryptic digested and analysed by MS. Graph shows the LFQ intensity of all the samples. *** *p* value < 0.01. (**B**) A fraction of the cell lysates of HEK293 cells used in A were blotted with the indicated antibodies. The whole Western blot figure can be found in Appendix A Original Blots and quantification for (**B**). (**C**) Venn diagram representation of the specific interactors identified in the MS-based proteomic screen in FLAG-KRAS WT (KRAS WT) or FLAG-KRASG12C (KRASG12C) immunoprecipitates treated with DMSO or 5 μM Sotorasib (i) for 24 h. (**D**,**E**) Graphs show average LFQ intensity of the indicated proteins in the different IPs. Error bars show SEM. * *p* value < 0.05, ** *p* value < 0.01.

**Figure 7 cancers-15-04141-f007:**
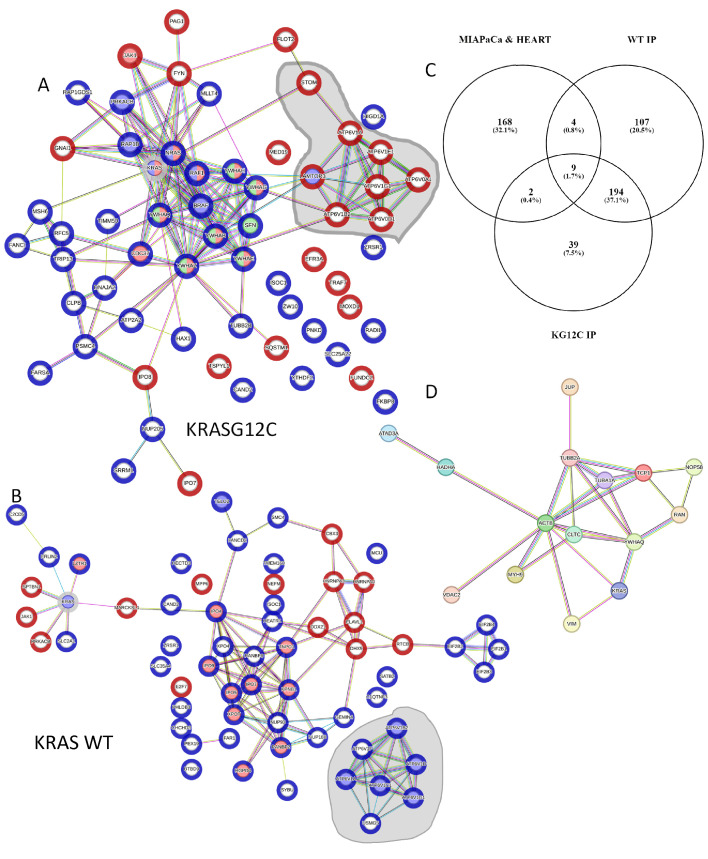
(**A**) Reconstructed network using StringDB of the dynamic interactome of KRASG12C (grey halo) in HEK293 cells identified in the AP-MS screen. Blue halo on proteins shows significantly increased binding to KRASG12C versus KRASG12C treated with Sotorasib *p* < 0.05. Red halos show significantly increased binding to KRASG12C treated with Sotorasib versus untreated KRASG12C *p* < 0.05. Proteins with blue nodes are involved in MAPK, proteins red nodes are PI3K related, and green nodes 14-3-3 domain superfamily related. Grey area shows proteins involved in lysosomal function and edges show evidence for interaction. (**B**) Reconstructed network using StringDB of the dynamic interactome of KRAS WT (grey halo) identified in the AP-MS screen. Blue halo on protein shows significantly increased binding to KRAS WT versus KRAS WT treated with Sotorasib *p* < 0.05. Red halo show significantly increased binding to KRAS WT treated with Sotorasib versus KRAS WT *p* < 0.05. Proteins with blue nodes are involved in mTOR signalling pathway, proteins red nodes are RAS GTPase binding. Edges show evidence for interaction. (**C**) Venn diagram representation of the proteins identified as specific interactors or KRAS WT and KRASG12C in our study with the proteins identified to be binding Sotorasib in MIAPaCa cells and heart tissue by Wang et al. [14]. (**D**) Network constructed using string DB of the 15 proteins that are in the intersection of the Venn diagram shown in (**C**). Edges show evidence for interaction (high confidence).

**Figure 8 cancers-15-04141-f008:**
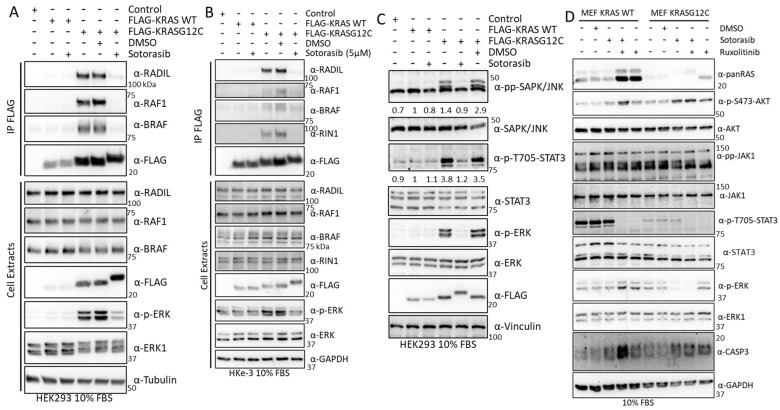
(**A**) HEK293 cells were transfected with the FLAG-KRAS WT or –KRASG12C; 24 h. Cells were treated with DMSO or 5 μM Sotorasib for 24 h. Cells were lysed and immunoprecipitated with FLAG antibody. IPs and cell extract were blotted with the indicated antibodies. (**B**) HKe-3 cells were transfected with the FLAG-KRAS WT or –KRASG12C. The cells were treated with DMSO or 5 μM Sotorasib for 24 h. Cells were lysed and immunoprecipitated with FLAG antibody. Proteins in IPs and cell extract were blotted with the indicated antibodies. (**C**) HEK293 cells treated as in A were blotted for the indicated antibodies. Numbers show fold changes in the level of phosphorylation of the indicated proteins. Intensities were measured using ImageJ. (**D**) MEF-KRAS WT and MEF- KRASG12C were treated with 5 μM Sotorasib and/or 5 μM Ruxolitinib as indicated for 24 h. Cells were lysed and the proteins were blotted with the indicated antibodies. The whole Western blot figure can be found in Appendix A Original Blots and quantification for Figure 8.

## Data Availability

The mass spectrometry proteomics data have been deposited to the ProteomeXchange Consortium via the PRIDE [75] partner repository with the dataset identifier PXD043536 and PDX043170.

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
