# Peer review of "Proteomic Mapping of the Interactome of KRAS Mutants Identifies New Features of RAS Signalling Networks and the Mechanism of Action of Sotorasib"

_cancers, 2023, doi:10.3390/cancers15164141_

Round 1

Reviewer 1 Report

The authors of manuscript no.: 2533304 provide results of a comprehensive proteomic study and validation of the interactome of different mutants of an important oncogene KRAS in vitro. Their data further extend the knowledge base about the previously described phenomenon of diverse cellular pathways affected by different KRAS mutants to varying extents with clinical consequences, e.g., tumor progression and resistance to targeted therapy. Importantly, the authors show that treatment of KRAS-12C mutant cells with inhibitor sotorasib changes KRAS interactome differently than in the wild type. This finding is relevant to resistance to sotorasib and adverse side effects of the treatment observed in the clinics. The study is very important for precision oncology, and its further validation may lead to clinically useful biomarkers and treatment decisions. It can also influence drug design. The manuscript is well-written and technically sound. There are a few points that should be considered in the revised version:

1/ The authors should provide more details about the rationale for the selection of human and murine cells for their experiments.

2/ Why the most frequent KRAS 12D variant has not been used for AP-MS (contrary to 12V, 12C, and 13D)?

3/ For transformation, migration, and survival experiments, different concentrations of siRNA-RADIL were used. What is the rationale for using the range between 4-20 nM?

4/ From Fig. 2 it is apparent that there are quite large differences in the studied phenomenon between different cell lines. The authors also admit that some target genes were not expressed in one cell line. I wonder whether the different genetic profiles of these models may explain some of these differences. For example, gene mutation, or large deletion or amplification of material on the chromosome where genes coding the studied proteins are located. Such profiles are available in databases, e.g., the COSMIC.

Minor comments:

- Page 10, line 439: …(Figure 4B and C not D).

English: Although I am not a native speaker, some minor typos or unclear sentences need correction, e.g.,

- word Graphs in the legend to Fig. 3

- sentence between lines 461-464 (network... interactors)

- line 626: ...screen

- sentences between lines 656-658 and 669-671 lack sense

-sentences between lines 656-658 and 669-671 lack sense

Reviewer 2 Report

Nolan et al

This well written manuscript describes an extensive body of work examining the interactome of wild type and 2 common mutants of K-RAS.  it includes studies to examine the effects of SotoRasib – a K-RAS 12C specific covalent inhibitor.  The study also identifies JAK as a potential new RAS signaling pathway.  Bearing in mind recent advances in the ability to target RAS proteins with small molecules and some of the unexpected results obtained with doing so, this type of study is important and is likely to be of considerable general interest.   The main weakness is consistent and that is a concern with apparent differences in RAS input levels in experiments, which might lead to a degree of over-interpretation of the data.

Comments

Minor

1.     The Simple Abstract may be a little too simple.

2.      There are a few spelling errors: line 431,  line 626,

3.     It is generally thought that RAF. PI3K and RALGDS are three of the main RAS mitogenic effectors.  Yet I do not see PI3K on the lists in the supplementary files, nor RALGDS.  Perhaps the authors could comment on this or highlight such interactions to make them a little more obvious.

4.     Figure 8 D: the most striking part of the figure is a strong increase in the levels of RAS in the presence of the JAK inhibitor.  This is mentioned in the discussion, but I wonder if the authors are prepared to speculate on a likely mechanism.   

Less Minor

1.     In Figure 1 the levels of K-RAS13D appear to be quite considerably lower than the other isoforms.  Can the reviewers discuss this and perhaps qualify their interpretation.

2.     Figure 2 there does not seem to be any quantification of RAS input.  Can the authors rule out the potential for different levels of RAS giving rise to different levels of interactor pull down, or explain why they feel this does not compromise their interpretation.

3.     In Figure 4 there are again some quite large differences in the relative inputs:   A: here seems to be very little SOS in the wtKRAS lane.  Does this affect the interpretation and is this a real effect or just experimental variation.  B:  The wild type KRAS input appears almost nil in the lysate but seems stronger in the IP.  Can the authors explain and discuss if this might affect the interpretation.  C:  wild type and 12D seem to be quite low compared to the other isoforms.  Does this affect the interpretation.  Likewise for D.

4.     Figure 5.  5/10uM of Sotorasib are quite high levels.  This drug works at nanomolar levels.  Do the authors feel this may have affected any results.

5.     Figure 8.    A:  wild type vs 12C RAS levels are quite large.  Do the authors feel this does not compromise their interpretation of the results.   
